Prediction of DNA binding proteins using local features and long-term dependencies with primary sequences based on deep learning

Li Guobin 1
Du Xiuquan dxqllp@163.com 2
Li Xinlu 1
Zou Le 1
Zhang Guanhong 1
Wu Zhize 1
1 School of Artificial Intelligence and Big Data, Hefei University , Hefei , China
2 School of Computer Science and Technology, Anhui University , Hefei , China
Uversky Vladimir
Electronic publication date: 2021 May 3
Publication date: 2021
Volume: 9
Electronic Location ID: e11262
Received 2020 Aug 31; Accepted 2021 Mar 22
Copyright: ©2021 Li et al.
Copyright year: 2021
Copyright holder: Li et al.
License: This is an open access article distributed under the terms of the Creative Commons Attribution License, which permits unrestricted use, distribution, reproduction and adaptation in any medium and for any purpose provided that it is properly attributed. For attribution, the original author(s), title, publication source (PeerJ) and either DOI or URL of the article must be cited.
License URL: https://creativecommons.org/licenses/by/4.0/

Keywords: DNA binding protein prediction, Deep learning, Convolution neural network (CNN), Long short-term memory network (LSTM), Long-term dependence, Fusion approach

Funding: National Natural Science Foundation of China 61672204 Key Scientific Research Foundation of Education Department of Anhui Province KJ2018A0555 Natural Science Foundation of Anhui Provincial 1908085MF184 University Natural Science Research Project of Anhui KJ2019A0835 Scientific Research and Development Fund of Hefei University 18ZR06ZDA 18ZR19ZDA 19ZR05ZDA This research was funded by the National Natural Science Foundation of China (61672204), the Key Scientific Research Foundation of Education Department of Anhui Province (KJ2018A0555), the Natural Science Foundation of Anhui Provincial (1908085MF184), the University Natural Science Research Project of Anhui (KJ2019A0835), the Scientific Research and Development Fund of Hefei University (18ZR06ZDA, 18ZR19ZDA, 19ZR05ZDA). The funders had no role in study design, data collection and analysis, decision to publish, or preparation of the manuscript.

==============================
DNA-binding proteins (DBPs) play pivotal roles in many biological functions such as alternative splicing, RNA editing, and methylation. Many traditional machine learning (ML) methods and deep learning (DL) methods have been proposed to predict DBPs. However, these methods either rely on manual feature extraction or fail to capture long-term dependencies in the DNA sequence. In this paper, we propose a method, called PDBP-Fusion, to identify DBPs based on the fusion of local features and long-term dependencies only from primary sequences. We utilize convolutional neural network (CNN) to learn local features and use bi-directional long-short term memory network (Bi-LSTM) to capture critical long-term dependencies in context. Besides, we perform feature extraction, model training, and model prediction simultaneously. The PDBP-Fusion approach can predict DBPs with 86.45% sensitivity, 79.13% specificity, 82.81% accuracy, and 0.661 MCC on the PDB14189 benchmark dataset. The MCC of our proposed methods has been increased by at least 9.1% compared to other advanced prediction models. Moreover, the PDBP-Fusion also gets superior performance and model robustness on the PDB2272 independent dataset. It demonstrates that the PDBP-Fusion can be used to predict DBPs from sequences accurately and effectively; the online server is at http://119.45.144.26:8080/PDBP-Fusion/.

Introduction

Protein-DNA interactions are widespread in all living organisms. A variety of biological processes are involved in these interactions including DNA replication, DNA repair, viral infection, DNA packaging, and DNA modification (Krajewska, 1992; Luscombe et al., 2000; Lou et al., 2014). In recent years, a larger number of DBP sequences have been generated by rapid advances in genomic and proteomic techniques. Exploring how protein-DNA interactions occur help us understand the genome.

In the early days, many experimental techniques have been proposed to predict DBPs. However, due to the time-consuming and money-consuming disadvantages of the experimental method, computational methods show great advantages in processing sequence data. So far, many effective computational prediction methods have been developed to identify DNA-binding proteins. They mainly use classic machine learning prediction models combined with complex feature extraction techniques. These methods include support vector machines (SVMs) (Cai & Lin, 2003; Bhardwaj et al., 2005; Yu et al., 2006; Kumar, Gromiha & Raghava, 2007; Zou, Gong & Li, 2013; Zhang et al., 2014; Liu et al., 2014; Fang et al., 2008), artificial neural networks (ANN) (Stawiski, Gregoret & Mandel-Gutfreund, 2003; Ofran, Mysore & Rost, 2007), decision tree (Tjong & Zhou, 2007), Bayesian network (Carson, Langlois & Lu, 2010), and Random forest (RF) (Wu et al., 2009; Kumar, Pugalenthi & Suganthan, 2009; Ma, Guo & Sun, 2016; Rahman et al., 2018). Support vector machines (SVM) and Random forest (RF) algorithm are widely used as classifiers for predicting DBPs and get better performance. Many sequence-based methods and web servers have been developed to identify DBPs. Recent methods and server names among them are: PseDNA-Pro (Liu et al., 2015), Local-DPP (Wei, Tang & Zou, 2017), SVM-PSSM-DT (Zaman et al., 2017), BindUP (Paz et al., 2016), PSFM-DBT (Zhang & Liu, 2017), HMMBinder (Zaman et al., 2017), iDNAProt-ES (Chowdhury, Shatabda & Dehzangi, 2017), DBPPred-PDSD (Ali et al., 2018), MSFBinder (Liu et al., 2018), DP-BINDER (Ali et al., 2019), and HMMPred (Sang et al., 2020).

In recent years, stack generalization (stack) as an integrated learning technique has gained much attention from researchers. StackDPPred (Mishra, Pokhrel & Hoque, 2019) first used features extracted from PSSM and residue-specific contact energy and then trained a stack-based machine learning method to predict DBPs. PredDBP-Stack (Wang et al., 2020) improved DBP prediction performance by exploring valuable features from the HMM profile. StackPDB (Zhang et al., 2020) took fusion features such as EDT, RPT, PseAAC, PsePSSM, and PSSM-TPC and then applied the stacked ensemble classifier to predict DBPs.

More and more evidence shows that it is practical to predict DBPs only from primary sequences. The traditional machine learning method shows superiority in solving the problem of small-scale data classification  (Bhardwaj et al., 2005; Yu et al., 2006). Unfortunately, these methods required the use of well-designed sequence features and evolutionary information features. They also need to be supported by relevant professional knowledge and experience. Additionally, feature extraction, training and forecasting cannot be performed simultaneously.

Recently, deep learning has been successfully applied to many big dataset classification tasks (Collobert et al., 2011; Krizhevsky, Sutskever & Hinton, 2017; Tayara, Soo & Chong, 2018; Tayara & Chong, 2018). Deep learning technology has incomparable advantages in the computation of large-scale DNA sequence data. For examples, Alipanahi and Delong first proposed DeepBind model based on deep learning technology to predict DNA binding proteins (Alipanahi et al., 2015). Zeng et al. (2016) predicted DNA binding sites based on CNN and many transcription factor data. They determined the best performance network structure by changing the CNN network width, depth, and pool operation design. Zhou et al. (2017) developed the CNNSite model based on a neural network and combined it with captured sequence features and evolution features to predict sequence binding residues. Shadab et al. proposed DeepDBP-ANN and DeepDBP-CNN models to identify DBPs. The former used a generated feature set trained on the traditional neural network. The latter used a previously learned embedding and a convolution neural network (Shadab et al., 2019).

Although several CNN based methods have been developed to predict DBPs, they are not good enough to achieve better accuracy in DBPs prediction. For instance, Zhang et al. (2019) developed the DeepSite model, which used Bi-LSTM and then CNN to capture long-term dependencies between the sequence motifs in DNA. Qu et al. (2017) used word embedding to encode sequences and then used CNN and LSTM as a classifier to predict DBPs. Hu et al. proposed CNN-BiLSTM method to identify DBPs. They coordinated a Bi-LSTM and a CNN (Hu, Ma & Wang, 2019) and also used word embedding technology. Recently, Du et al. (2019) developed a deep learning method named MsDBP, which obtained four-vectors (multi-scale features) based on 25%, 50%, 75%, and 100% of the original sequence length. Moreover, they used many dense layers to learn different abstract features for predicting DBPs  (Du et al., 2019).

These methods based on CNN can only represent the local dependence of the DNA sequence information, but the long-term dependence of the sequence is not considered. It is a better choice to predict DBPs by fusing the local features obtained by CNN with the long-term context-dependent features captured by Bi-LSTM. Although some existing methods also combined CNN and LSTM to predict DBPs, they both completed the preprocessing of gene sequences based on word embedding. The One-hot encoding method used in this paper is simple and effective, and the pre-processing is more efficient. In addition, we also need to consider the design of a convolutional neural network with the size, characteristics, and distribution of the sequence data.

In this study, we develop the PDBP-Fusion method to overcome the disadvantages of the existing methods. One-hot encoding was used in this approach, which is easier and faster than word embedding. Moreover, CNN was used to obtain the local features of DNA sequences through self-learning, and Bi-LSTM was used to capture long-term dependencies in the sequence context. Finally, a fusion feature combining local and global characteristics was used to predict DBPs. The contributions of this research are described as follows

(1) Since the length of the DNA sequence variation, the optimal sequence length is determined using experiments to obtain the best sequence characteristics. In this article, a grid search method based on the sequence length distribution is suggested to search for the best truncated sequence length parameters.

(2) A new method, PDBP-Fusion, has been developed based on deep learning to predict DBPs. It consists of a CNN network and a Bi-LSTM network. The former is responsible for extracting the abstract features layer by layer. Simultaneously, the latter is responsible for obtaining long-term dependencies in the sequence context.

(3) The proposed method does not require manual extraction of data features; it uses only deep learning to self-learn original sequence features based on the primary sequence. Two coding practices, One-hot and word embedding, were used to predict DNA binding sites. The optimal network structure was found by a parameter grid-search strategy based on the benchmark data set for predicting DBPs.

Materials & Methods

In this section, we first introduce the benchmark dataset and the independent test dataset. Next, we describe our proposed PDBP-Fusion framework. Finally, we illustrate all the details concerning the construction of the model, the evaluation of the model and the experimental setting of the parameters.

Datasets

We use the benchmark dataset obtained from Ma, Guo & Sun (2016) referred to as PDB14189. The PDB14189 dataset is composed of 7,129 DBPs (positive samples) and 7,060 non-DBPs (negative samples). All of them are from the UniProt (Apweiler et al., 2012) database. This dataset is identical to MsDBPs (Du et al., 2019).

In addition, we used an independent test dataset, PDB2272, to compare the performance of our proposed model with other existing prediction methods (Rahman et al., 2018; Liu et al., 2015; Wei, Tang & Zou, 2017; Du et al., 2019). We obtained original dataset consisting of 1,153 DBPs and 1,153 non-DBPs from Swiss-Prot. We removed sequences that had more than 25% similarity and filtered out sequences with irregular characters (“X” or “Z”). Finally, the PDB2272 dataset contained 1,153 DBPs and 1,119 non-DBPs.

Framework of the PDBP-fusion model

In this study, we develop a deep learning model called PDBP-Fusion, which combined CNN and Bi-LSTM, to predict DNA binding proteins. The former obtained local DNA sequences through self-learning and the latter learned the long-term dependencies in the sequence context. The proposed models consist of the sequence encoding layer (one-hot encoding or embedding encoding), the local feature learning layer, the long-term context learning layer, and the synthetic prediction layer. Figure 1 illustrates the main framework of the PDBP-Fusion model.

Figure 1 Architecture of the proposed PDBP-Fusion model.

Sequence encoding

Feature coding is a critical task in building machine learning models. And it is preferable to obtain a suitable coding scheme after observing the characteristics of the dataset. The statistical results showed that the sequence length of the DNA benchmark database ranged from 50 to 2,743. The sequence length distribution is provided in Fig. 2.

Figure 2 Statistical graph of DNA sequence length distribution in the PDB14189 dataset.

Since the sample length of the DNA sequence varies, it is necessary to choose an appropriate maximum sequence length for data processing. A maximum length was chosen as the best experimental endpoint reference for the PDBP-Fusion model, as determined by a series of experiments between 100 and 1,000. Further details on the results of the comparison experiment are available in section 3.2.

(a) One-hot encoding

One-hot encoding is a general method that can vectorize any categorical features. In the One-hot encoding method, amino acid must be encoded numerically.

For example, a DNA sequence “S= EFDYVICEEE” was taken from Fig. 3A and encoded with the One-hot approach. An output vector, with a dimension of 21*d, was obtained from a word embedding encoding of the input sequence S = {S0, S1, S2, …Sd}.

Figure 3 Coding diagram of (A) One-hot encoding and (B) word embedding encoding.

(b) Embedding encoding

Embedding is used to represent discrete variables as continuous vectors. It produces a dense vector with a fixed, arbitrary number of dimensions. Word embedding is one of the most popular representation of document vocabulary. The word embedding (Collobert et al., 2011) representation can reveal many hidden relationships between phrases.

When using word embedding, the input sequence is converted into a numerical code based on Table 1, and the digitally encoded protein sequence is converted into dense vectors.

Table 1 Amino acid encoder.

Amino acids	Encode (“A:1” denote “encode A with number 1”)	
A∼Z (except B, J, O, U, X, Z)	A:1, C:2, D:3, E:4, F:5, G:6, H:7, I:8, K:9, L:10, M:11, N:12, P:13, Q:14, R:15, S:16, T:17, V:18, W:19, Y:20	
B, J, O, U, X, Z:0	

During this period, the sequence encoding layer generates a fixed-length feature represented by Encode1 from the DNA binding protein sequence, using One-hot encoding or Word embedding encoding. (1) Encode1=EncodeS0,S1,S2,…Sn.

Local feature learning

A convolutional neural network was utilized to detect the functional domains of protein sequences. The local feature-leaning layer consisted of several blocks that performed convolution, batch-normalization, ReLU, and max-polling operation. Figure 4 presents the concrete structure.

Figure 4 CNN network structure.

This layer can use a One-hot encoding or word embedding encoding approach before the CNN structure. The specific network structure and parameter are discussed in section 2.6. Experimental results using different encoding methods are presented in section 3.1, 3.2 and 3.3.

The local feature learning layer generates a representation of fixed length features which can be designated as Local2. (2) Local2=Localencode0,encode1,encode2,…encoden.

Long-term context learning

CNN-based prediction methods can get only the local characteristics of gene sequences. Since the gene sequence is long enough, it is desirable to use BI-LSTM to identify long-term dependencies. In our proposed model, the long-term context learning layer results in a characteristic representation of a fixed length, designated by Long_term3. (3) Long_term3=Long_termlocal0,local1,local2,…localn.

Synthetic prediction

The entry of the previous layer was concatenated into a vector and then went through a fully connected layer. Next, the hidden neurons “vote” on each of the labels, and the winner of that vote is the classification decision. The sigmoid function was used as the network activation function and cross-entropy function as the loss function. The fully connected layer generates output represented by Sout. (4) Sout=Syntheticlong_term0,long_term1,long_term2,…long_termn.

Model construction and evaluation

Several validation methods were used to evaluate the performance of the proposed models. In a series of publications (Krajewska, 1992; Luscombe et al., 2000; Lou et al., 2014; Cai & Lin, 2003; Bhardwaj et al., 2005; Yu et al., 2006) in the field of Bioinformatics, k-fold cross-validation was widely used. In this paper, all experiments used 5-fold cross-validation to assess model performance on the PDB14189 dataset. Due to the relatively broad fluctuation range of the prediction results based on deep learning, the k-fold (k = 5) cross-validation was repeated five times. The average value was used in assessing the performance of the model. When evaluating model performance on the PDB14189 dataset, we follow the steps which illustrated in Fig. 5.

Figure 5 Model evaluation on benchmark datasets PDB14189.

Step 1. Take 11,351 samples as the training set and take 2,838 as the test set.

Step 2. Divide the 11,351 samples into two sections: (1) 10,215 samples were used for training, and (2) the remaining 1,136 samples were used for verification.

Step 3. Repeat k-fold (k = 5) cross-validation five times. The mean value was used to measure the performance of the model.

We conduct the independent test on the PDB2272 dataset as follows

Step 1. Train the PDBP-Fusion model, which take 80% of the samples in PDB14189 as a training set and use the rest (20%) as a validation set.

Step 2. Save the well-trained PDBP-Fusion model with the optimal parameter configuration, and then predict the PDB2272 independent dataset.

Step 3. Compare the prediction results with other methods to evaluate model generalization.

Five evaluation indicators, including accuracy (ACC), precision (PRE), sensitivity (SE), specificity (SP), Matthew’s Correlation Coefficient (MCC), were used as the performance measure. The various performance measures were defined as follows

(5) ACC=TP+TNTP+NP+FP+FN

(6) PRE=TPTP+FP

(7) SE=TPTP+FN

(8) SP=TNTN+FP

(9) MCC=TP*TN−FP*FNTP+FP∗TP+FN∗TN+FP∗TN+FN

TN, FN, TP, and FP represent the number of true negative, false negative, true positive, and false positive samples predicted. The area under the ROC (AUC) (Fawcett, 2004) is also used to evaluate prediction performance.

Experimental parameter configuration

The entire procedure was implemented based on the Keras framework. Complete codes, including the One-hot code, word-embedding code, CNN, Bi-LSTM, PDBP-CNN, and PDBP-Fusion code, are provided via http://119.45.144.26:8080/PDBP-Fusion/. Table 2 gives the detailed parameters of the proposed models.

Table 2 Parameters details of the proposed models.

Layers	PDBP-CNN	Output shape	PDBP-Fusion	Output shape	
1	One-hot encoding	Len*20	One-hot encoding	Len *20	
2	Convolution1 (kernel=7, stride=1)	Len *64	Convolution1 (kernel=9, stride=1)	Len *64	
3	Max-polling1 (kernel=2)	(Len / 2) *64	Max-polling1 (kernel=2)	(Len / 2) *64	
4	Convolution2 (kernel=7, stride=1)	(Len / 2) *64	Convolution2 (kernel=9, stride=1)	(Len / 2) *64	
5	Max-polling2 (kernel=2)	(Len / 4) *64	Max-polling2 (kernel=2)	(Len / 4) *64	
6	Convolution3 (kernel=7, stride=1)	(Len / 2) *64	Bi- LSTM(32)	150*64	
7	Max-polling3 (kernel=2)	(Len / 8) *64	Dense(128)	128	
8	Dense (128)	128	Dense(2)	2	
9	Dense (2)	2			
Notes.

“Len” denotes the input sequence max length.

Results

In this section, we first elaborate two series of comparative experiments based on different lengths of the sequence dataset. Next, we select other model parameters such as dropout ratio and convolution kernel size to obtain the optimal parameter configuration of the PDBP-Fusion model. Finally, we present the performance of PDBP-Fusion with other published studies on the same benchmark dataset PDB14189 and the independent dataset PDB2272.

Performance comparison of PDBP-CNN models using One-hot encoding

During the data processing phase, we selected different max length from 500 to 1,000 to encode the DNA sequence. Then we evaluate the overall performance of the PDBP-CNN model based on One-hot encoding. When the maximum sequence length exceeded 1,000, it became impossible to finish the 5-fold cross-validation experiment five times due to excessive memory consumption. Table 3 shows the experimental results.

Table 3 Quantitative results of the PDB-CNN method with different maximum sequence lengths.

Lmax	ACC (%)	SE (%)	PRE (%)	SP (%)	MCC (%)	AUC (%)	
100	76.51	73.83	83.53	69.43	54.05	85.26	
150	78.85	76.92	83.29	74.37	58.27	87.21	
200	79.48	77.9	83.28	75.64	59.55	88.02	
250	80.38	78.61	84.17	76.56	61.21	88.71	
300	80.72	79.42	83.70	77.71	61.90	89.06	
350	81.04	77.95	87.32	74.71	62.86	89.42	
400	81.28	78.34	87.39	75.11	63.40	89.7	
450	81.52	79.21	86.20	76.80	63.59	89.94	
500	81.55	78.72	87.36	75.68	63.89	90.00	
550	81.32	78.52	87.21	75.38	63.46	89.86	
600	81.88	80.64	84.67	79.05	64.21	90.18	
650	81.74	80.58	84.49	78.97	63.97	90.12	
700	81.79	78.22	88.83	74.68	64.42	90.13	
750	81.68	79.04	87.07	76.24	64.08	89.92	
800	81.71	78.33	88.44	74.92	64.28	90.11	
850	82.02	79.25	87.49	76.50	64.69	90.29	
900	81.94	79.09	87.51	76.32	64.53	90.13	
950	81.91	79.50	86.74	77.03	64.44	90.24	
1000	82.04	80.33	85.40	78.65	64.43	90.05	
>1000	–	–	–	–	–	–	

Table 3 shows that convolutional neural networks can learn sophisticated features. The best performance (MCC = 64.69% and ROC-AUC = 90.29%) was achieved with a sequence length equals 850. However, model performance did not increase monotonically as maximum sequence lengths increased. The experimental results showed little difference when the maximum sequence length exceeds 700 under the same structural network model.

Performance comparison of PDBP-Fusion models using One-hot encoding

Previously, the classic CNN network model was used for prediction, which does not obtain the long-term context dependencies in sequences. This subsection describes the use of PDBP-Fusion to evaluate performance. Table 4 presents the comparative experimental results.

Table 4 Quantitative results of the PDBP-Fusion method with different maximum sequence lengths.

Lmax	ACC (%)	SE (%)	PRE (%)	SP (%)	MCC (%)	AUC (%)	
100	77.28	74.53	83.99	70.50	55.42	85.79	
150	78.91	76.62	84.24	73.53	58.57	87.36	
200	80.08	77.88	84.77	75.35	60.71	88.33	
250	80.74	79.51	83.56	77.89	61.9	88.96	
300	81.44	79.57	85.25	77.60	63.33	89.50	
350	82.27	80.39	85.8	78.71	64.83	90.01	
400	81.70	79.48	86.38	76.98	64.12	90.11	
450	82.30	79.36	87.84	76.71	65.15	90.37	
500	82.50	79.58	87.95	77.01	65.54	90.43	
550	82.16	79.97	86.56	77.72	64.90	90.37	
600	82.56	80.87	85.90	79.18	65.50	90.61	
650	82.81	80.84	86.56	79.03	66.02	90.7	
700	82.81	81.02	86.45	79.13	66.1	90.83	
750	82.66	80.51	86.82	78.45	65.8	90.65	
800	82.7	79.99	87.8	77.56	65.95	90.74	
850	82.71	80.51	86.93	78.44	65.89	90.73	
900	82.63	80.06	87.61	77.61	65.85	90.69	
>900	–	–	–	–	–	–	

In this series of experiments, the best performances (MCC of 66.1% and ROC-AUC of 90.83%) were achieved. Experience has shown that the performance of the PDBP-Fusion model does not improve with increasing sequence length. When the sequence length is 700, the optimum performance was achieved, after which it gradually decreased. Experiments show that the Bi-LSTM network can capture long-term dependencies even with a sequence length of less than 700.

Performance comparison of PDBP-Fusion models using word embedding

In this section, we reported the model performances of PDBP-CNN and PDBP-Fusion based on word embedding encoding. We conducted two identical experiments with different sequence lengths that vary from 100 to 1,000. After the sequence length exceeded 1,000, it became impossible, as before, to complete the 5-fold cross-validation five times due to excessive GPU memory consumption. The best performances listed in Table 5 are each presented with their two best-archived results for all sequence lengths. Experiments have shown that the PDBP-Fusion approach can obtain all-round performance advantages superior to the PDBP-CNN method based on word embedding encoding.

Table 5 PDBP-Fusion model performance using a word embedding encoding on the PDB14189 dataset.

Methods	ACC (%)	SE (%)	SP (%)	MCC (%)	AUC (%)	
PDBP-Fusiona	81.01	78.48	81.58	62.0	89.03	
PDBP-Fusionb	79.40	83.60	75.15	59.1	87.81	
Notes.

a PDBP-Fusion model: (length = 800, word embedding encoding, 64 convolution kernels).

b PDBP-Fusion model: (length = 800, word embedding encoding, 32 convolution kernels).

Model parameter selection and optimization

Based on the comparative experiments in section 3.1, 3.2 and 3.3, it is apparent that PDBP-CNN and PDBP-Fusion with the One-hot encoding approach showed better performance than the word embedding practice. Figure 6 shows that the PDBP-Fusion model with One-hot encoding obtained the best results (MCC = 66.10%, AUC = 90.83%) and that the PDBP-CNN model with one-hot encoding obtained the second-best performance result (MCC = 64.69%, AUC = 90.29%).

Figure 6 MCCs and AUCs of the top three proposed models.

The best-performing architectures were then identified by varying the CNN convolution kernel width and the dropout ratio, as described in the following sections. The previous two best model performances were improved by tuning the network parameters.

Selecting different dropout ratio s in CNN

As shown in Fig. 7, the variation range of dropout parameters of the PDBP-Fusion model was [0.1,0.2,0.3,0.4], whereas the other parameters remained unchanged. MCC and AUC both reached their optimal values when dropout ratio = 0.3. In the same case, the optimal performance of the PDBP-CNN model was achieved when dropout ratio = 0.2.

Figure 7 MCCs and AUCs of models with different dropout ratios (violin plot).

Selecting different convolution kernels in CNN

As shown in Fig. 8, the convolution kernel size parameter of the PDBP-Fusion model varies in the range of [5,7,9], whereas the other parameters remained unchanged. The model achieved optimal performance (MCC = 66.10%, AUC = 90.83%) when the convolution kernel size = 9. In the same case, the optimal performance of the PDBP-CNN model was reached when the convolution kernel size = 7.

Figure 8 MCCs and AUCs of models with different dropout rates (box plot).

The optimal models for PDBP-CNN and PDBP-Fusion were identified based on a series of experiments. Table 6 gives details of these performance results. The encoding approach and the network design parameters are listed for each.

Table 6 Peak performance of PDBP-CNN and PDBP-Fusion models on the PDB14189 dataset.

Methods	ACC (%)	SE (%)	SP (%)	MCC (%)	AUC (%)	
PDBP-CNNa	82.02 ± 1.22	87.49 ± 4.12	76.50 ± 5.66	64.69 ± 1.87	90.29 ± 0.51	
PDBP-Fusionb	82.81 ± 1.30	86.45 ± 4.59	79.13 ± 5.81	66.1 ± 2.04	90.83 ± 0.57	
Notes.

a PDBP-CNN model: The maximum length is 850. The convolution layer has three layers, the convolution kernel is (7*1), the maximum pooling size is (2,1), and the total connection layer has 128 nodes. The dropout rate is set to 0.2.

b PDBP-Fusion model: The maximum length is 700. The convolution layer has two layers, the convolution kernel is (9*1), the maximum pooling size is (2,1), the number of cells in Bi-LSTM is set to 16*2, and the total connection layer has 128 nodes. The dropout rate is set to 0.3.

Performance comparison on the benchmark dataset

In this section, we compare the performance of PDBP-Fusion with previously published methods such as DNABP (Ma, Guo & Sun, 2016), MsDBP (Du et al., 2019) and StackDPPred (Mishra, Pokhrel & Hoque, 2019) approach on the same benchmark dataset. The DNABP approach (Ma, Guo & Sun, 2016) combines various carefully selected manual features beyond the scope of this work. They rely on biological databases and require biological expertise. The PDBP-CNN and PDBP-Fusion methods are based only on the primary sequence and do not require manual feature extraction.

A comparison experiment based on StackDPPred (Mishra, Pokhrel & Hoque, 2019) and One-hot encoding were carried out on the PDB14189 dataset. We first use One-hot for encoding an input sequence and then flatten the input vector. We entered these features into StackDPPred method and explored Base and Meta Classifiers. In order to find the base-classifiers to use in the first stage and the meta-classifier to use in the second stage of stacking framework, four different machine learning algorithms such as SVM, KNN, LogReg and RDF were explored. Figure 9 shows the StackDPPred prediction framework using One-hot encoding.

Figure 9 Overview of the StackDPPred prediction framework based on One-hot encoding.

We optimize each classifier by cross-validation based on 50% of the PDB14189 dataset. And we evaluate the trained model’s performance on the remaining 50% of the PDB14189.

(i) SVM: The best values of the base-classifier SVM parameters are C = 1 and γ = 0.0001. Likewise, the best values of the parameters of the SVM, used as meta-classifier, are C = 1 and γ = 0.0001.

(ii) Logreg: In our implementation, we find C = 0.0400 results in the best accuracy.

(iii) RDF: In our implementation of the RDF ensemble learner, we have used bootstrap samples to construct 2,000 trees in the forest.

(iv) KNN: In this work, the value of k is set to 6, and all neighbours are weighted uniformly.

Table 7 indicates that the PDBP-Fusion methods achieved better performance than some random forest classifier models (Ma, Guo & Sun, 2016) with manually extracted features such as PSSM, PSSM-PP, PHY, etc. Our approach used the characteristics of deep learning and self-learning ability to identify DBPs based only on the sequences. Experimental results show that the performance of the PDBP-Fusion method was significantly improved compared with that of MsDBP. The MCC values for PDB-CNN and PDB-Fusion increased by at least 9.1% and 6.7% respectively. The AUC values increased by 2.8% and 2.2%, respectively.

Table 7 Comparison of the proposed model with other methods on the PDB14189 dataset.

Methods	ACC (%)	SE (%)	SP (%)	MCC (%)	AUC (%)	
MsDBP	80.29	80.87	79.72	60.61	88.31	
PSSMa	79.62	76.02	83.21	59.4	–	
PSSM-PPa	81.69	78.92	84.45	63.5	–	
PHYa	77.65	73.54	81.76	55.5	–	
PSSM-PP+BP_NBPa	83.68	81.01	86.34	67.4	–	
PSSM-PP+PHYa	82.67	79.95	85.39	65.4	–	
BP ± NBP+PHYa	80.40	76.88	83.92	60.9	–	
ALL featuresa	84.64	82.23	87.06	70.6	–	
64 Optimal featuresa	86.90	83.76	90.03	72.7	–	
StackDPPred(One-hot)b	76.00	79.27	72.71	52.10	83.18	
PDBP-CNN	82.02	87.49	76.50	64.69	90.29	
PDBP-Fusion	82.81	86.45	79.13	66.1	90.83	
Notes.

a DNABP method which using RF classifier and various features (Ma, Guo & Sun, 2016).

b StackDPPred(One-hot) method using StackDPPred and One-hot encoding (Mishra, Pokhrel & Hoque, 2019).

Performance comparison on the independent test datasets

The PDBP-Fusion model was then evaluated on the independent dataset (PDB2272) to verify its robustness. The comparison covered the proposed method with other advanced methods. Table 8 shows the experimental results.

Table 8 Comparison of various machine learning methods on the PDB2272 dataset.

Methods	ACC (%)	SE (%)	SP (%)	MCC (%)	AUC (%)	
Qu et al. (Qu et al., 2017)	48.33	49.07	48.31	−3.34	47.76	
Local-DPP (Wei, Tang & Zou, 2017)	50.57	58.72	8.76	4.56	–	
PseDNA-Pro (Liu et al., 2015)	61.88	59.90	75.28	24.30	–	
DPP-PseAAC (Rahman et al., 2018)	58.10	59.10	56.63	16.25	61.00	
MsDBP (Du et al., 2019)	66.99	66.42	70.69	33.97	73.83	
PDBP-Fusion	77.77	73.31	66.85	56.65	85.39	

In Table 8, the ACC value of PDBP-Fusion on PDB2272 exceeds other prediction methods. The ACC of PDBP-Fusion is 77.77%, which is 16.7% higher than the ACC of MsDBP (77.77% vs 66.99%). From the perspective of model stability, the MCC of PDBP-Fusion is 0.5665, which is 66.8% higher than the MCC of MsDBP. It demonstrates that the PDBP-Fusion model has got superior performance and model robustness on the PDB2272 independent dataset.

Web server for PDBP-Fusion

Many advanced methods (Chen et al., 2017; Qiu et al., 2018; Cheng et al., 2019; Chou, Cheng & Xiao, 2019) provide an available Web server and prediction tool for users to predict DBPs online. We also offer a Web server at http://119.45.144.26:8080/PDBP-Fusion/. Additionally, we provide all the steps to get the predicted results for convenience.

Figure 10 Index page of the web server.

Step 1. Click the link, and you will see the index page is shown in Fig. 10.

Step 2. Click the “Download” link, and you can download the benchmark dataset, independent dataset, and the codes.

Step 3. Either type or copy and paste the protein sequence into the input box in Fig. 10, Click the “Predict” button to see the predicted results.

Conclusions

The CNN based method alone is not accurate enough in predicting DBPs from DNA sequences. In this his study, the CNN network was used to find suitable local features, and Bi-LSTM was used to capture long-term dependencies among DNA sequences. It is preferable to predict DBPs by merging the local features obtained by CNN with the long-term context-dependent features captured by Bi-LSTM. Some existing methods have also combined CNN and LSTM to predict DBPs, but they completed the pre-processing of gene sequences based on word embedding. The One-hot encoding method used in this study is straightforward and more efficient than word embedding. It is also necessary to consider designing a convolutional neural network reasonably to match the DNA sequence data’s characteristics and distribution.

The PDBP-Fusion method proposed in this paper has demonstrated its significance on the PDB14189 benchmark dataset and its performance relative to the outcomes of existing methods on the PDB2272 independent dataset. The proposed method showed remarkably higher generalization ability compared with existing methods. Furthermore, this study suggests that the fusion approach, combining local features and long-term dependencies, will be necessary for sequence-based tasks in genomics. It also provides a solution to other sequential prediction problems.

Additional Information and Declarations

Competing Interests

Author Contributions

Data Availability

The authors declare there are no competing interests.

Guobin Li conceived and designed the experiments, performed the experiments, prepared figures and/or tables, and approved the final draft.

Xiuquan Du conceived and designed the experiments, performed the experiments, prepared figures and/or tables, authored or reviewed drafts of the paper, and approved the final draft.

Xinlu Li performed the experiments, prepared figures and/or tables, and approved the final draft.

Le Zou, Guanhong Zhang and Zhize Wu analyzed the data, authored or reviewed drafts of the paper, and approved the final draft.

The following information was supplied regarding data availability:

The datasets and source codes for this study are freely available to the academic community at GitHub: https://github.com/hfuulgb/PDBP-Fusion.

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
