# Peer review of "Prediction of DNA binding proteins using local features and long-term dependencies with primary sequences based on deep learning"

_PeerJ, doi:10.7717/peerj.11262_

## Round 0.1 · original submission · Major Revisions

As you can see, the recommendations of the reviewers are as diverse as they could be, ranging from reject to minor revision. However, this diversity gives enough room for improvement. Please address all the critiques raised by the reviewers and revise your manuscript accordingly.

Reviewer 1 ·

Basic reporting

Prediction of DNA-binding protein is a hot topic in protein sciences. Many researchers worked and published a series of papers using diverse methods. In the current study, the authors proposed the PDBP-Fusion method based on deep learning (DL) to identify DBPs using primary sequences. The current manuscript can be published after covering the following concerns.
Comment 1: In line 169, keep space before “Accuracy”.
Comment 2: In line 232, text “in the follows” is not clear. Please provide alternate text. Similarly, replace “. And we” with “and”.
Comment 3: In line 265, keep space before “Therefore”.
Comment 4: In line 272, write “employed” after “we”.

Experimental design

Comment 1: The authors only used one training dataset. To evaluate the generalization power of the proposed method, authors must use the independent dataset i.e., PDB2272 already employed by Du. X et al. in their method (MsDBP: Exploring DNA-binding Proteins by Integrating Multi-scale Sequence Information via Chou’s 5-steps Rule).

Comment 2: Kindly compare the independent dataset (PDB2272) results with Du. X et al. study and include the comparison discussion in the manuscript.

Comment 3: The manuscript should be in proper structure. Section 3.1. Datasets need to discuss in Material and Method section.

Comment 4: The independent dataset (PDB2272) should be explained in Datasets section before section 2.1

Validity of the findings

Comment 1: The authors did not provide the codes for the proposed method. To reproduced/validate the proposed predictor, authors should be provide complete code i.e. one-hot code, word-embedding code, CNN and BiLSTM code, PDBP-CNN and PDBP-Fusion code, Figure 2, 5, 6 codes.
Comment 2: The authors used the fixed length sequences such as 600, 800. It is possible that important information may be lost. Authors should perform experiments on original length sequences and explain the obtain results in the manuscript.
Comment 3: In line 110, authors stated “we manually removed the sequence length less than 50” What is the number of sequences in the remaining dataset? If the number of sequences reduced in the dataset, how authors claim a fair comparison with the existing predictors.
Comment 4: Authors missed recently developed machine learning methods such as DBPPred-PDSD, DP‑BINDER, StackPDB, PredDBP-Stack for prediction of DNA binding proteins in the literature. Please discuss in the introduction section.

Comment 5: “To assess the performance of a proposed model, several validation methods are adopted. In a series of publications [1-6] in the field of bioinformatics, k-fold cross validation was widely used.”. Please add these lines in section 3.3 Model training and evaluation.

Additional comments

Comment 1: Several mistakes have been found. Please read the whole manuscript for spelling mistakes, grammar mistakes, and sentence structures.

Reviewer 2 ·

Basic reporting

The paper is well written and gives a clear understanding of the idea of using deep learning to predict DNA binding proteins. The figures are clearly annotated.

Experimental design

The research idea is well defined in the literature review and there is substantial evidence in the results to prove the validity of the experimental design. I found a method missing in the comparison of results.

1. I believe the method I have attached here has great performance on the PDB dataset mentioned in their paper. StackDPPred: a stacking based prediction of DNA-binding protein from sequence has great performance and they also have a local copy of the software. I would like the author to compare the results for this method on the dataset PDB14189. I would highly recommend adding this method in comparison to Table 7 in the manuscript.
2. In Table 3 and Table 4, the caption should be rewritten.
3. I highly recommend the author to add an independent test dataset to validate the findings of the research.
4. The webserver provided doesn’t work. Please fix this.

Validity of the findings

The findings in the paper are slightly better than the methods previously published by the researchers. Conclusions are well stated in terms of answering the question about DNA binding protein prediction. As mentioned above I speculate the performance of the predictor as compared to the StackDPPred. I would highly recommend the comparison of the PDBP-Fusion with StackDPPred.
As soon as the performance of this research outperforms the StackDPPred, I believe the novelty of the method can be validated.

Additional comments

Paper is well structured and clearly written.

Reviewer 3 ·

Basic reporting

The authors use deep learning methods for the prediction of DNA binding proteins from sequences. However, significant changes must be made to experimental design to justify its relevance (like comparing with other state-of-the-art methods) on test datasets.

Experimental design

Methods are old and hence not appropriate as well as not well described. There are several latest papers with very good results which are not being compared or acknowledged.

Validity of the findings

The work does not include the necessary controls.

As mentioned above, the study does not include a test dataset. Also, the authors mention DNABP on the dataset section which has a different number of sequences than the ones the authors use.

Additional comments

1) The authors present a method for prediction of DNA binding proteins from primary sequences alone (without PSSM) and have comparable performance to other machine learning methods on the dataset explored.

2) Ma et. al have a different numbers of sequences in the benchmark dataset and test dataset compared to MsDBP. The authors say both datasets are similar but we didn't find it to be true.

3) The authors perform 5 fold cross-validation without a validation set and don't have a test dataset. The authors are probably aware that deep learning methods overfit easily and there should be a validation/test set at minimum to claim good performance.

---

## Round 0.2 · accepted · Accept

All critiques were addressed and the manuscript was revised accordingly. Therefore revised version is acceptable now.

Reviewer 2 ·

Basic reporting

I believe the revised manuscript addresses all the concerns mentioned in the review. The implementation outperforms the previous methods and is a great addition in the field of DNA binding protein prediction.

Experimental design

The experimental design is concise and well thought.

Validity of the findings

The findings can be validated through the PDB14189 dataset results.